# High-Risk Human Papillomavirus and Tobacco Smoke Interactions in Epithelial Carcinogenesis

**DOI:** 10.3390/cancers12082201

**Published:** 2020-08-06

**Authors:** Francisco Aguayo, Juan P. Muñoz, Francisco Perez-Dominguez, Diego Carrillo-Beltrán, Carolina Oliva, Gloria M. Calaf, Rances Blanco, Daniela Nuñez-Acurio

**Affiliations:** 1Universidad de Tarapacá, Arica 1000000, Chile; 2Advanced Center for Chronic Diseases (ACCDiS), Facultad de Medicina, Universidad de Chile, Santiago 8330024, Chile; 3Instituto de Alta Investigación, Universidad de Tarapacá, Arica 1000000, Chile; juanpablomunozbarrera@gmail.com (J.P.M.); gmc24@cumc.columbia.edu (G.M.C.); 4Laboratorio Oncovirología, Programa de Virología, Instituto de Ciencias Biomédicas, Facultad de Medicina, Universidad de Chile, Santiago 8380000, Chile; francisco.perez.d@ug.uchile.cl (F.P.-D.); diegocb17@hotmail.com (D.C.-B.); carolinaolivaguerrero@gmail.com (C.O.); rancesblanco1976@gmail.com (R.B.); danynuac@gmail.com (D.N.-A.); 5Center for Radiological Research, Columbia University Medical Center, New York, NY 10032, USA

**Keywords:** papillomavirus, tobacco, smoking, cancer, cervix

## Abstract

Cervical, anogenital, and some head and neck cancers (HNC) are etiologically associated with high-risk human papillomavirus (HR-HPV) infection, even though additional cofactors are necessary. Epidemiological studies have established that tobacco smoke (TS) is a cofactor for cervical carcinogenesis because women who smoke are more susceptible to cervical cancer when compared to non-smokers. Even though such a relationship has not been established in HPV-related HNC, a group of HPV positive patients with this malignancy are smokers. TS is a complex mixture of more than 4500 chemical compounds and approximately 60 of them show oncogenic properties such as benzo[α]pyrene (BaP) and nitrosamines, among others. Some of these compounds have been evaluated for carcinogenesis through experimental settings in collaboration with HR-HPV. Here, we conducted a comprehensive review of the suggested molecular mechanisms involved in cooperation with both HR-HPV and TS for epithelial carcinogenesis. Furthermore, we propose interaction models in which TS collaborates with HR-HPV to promote epithelial cancer initiation, promotion, and progression. More studies are warranted to clarify interactions between oncogenic viruses and chemical or physical environmental factors for epithelial carcinogenesis.

## 1. Introduction

Cervical cancer is a malignant neoplasia that develops in the cervix, the lower part of the uterus that connects this with the vagina [1]. This disease is initiated through a precursor epithelial lesion with slow and progressive evolution, known as cervical intraepithelial neoplasia (CIN I, II, and III). When only stratified epithelium is affected, it constitutes an in-situ cervical carcinoma. Eventually, the progression to invasive cancer occurs when tumor cells invade additional tissues [2]. Human papillomavirus (HPV) infection is the necessary condition for developing cervical carcinoma, because almost 100% of patients with cervical cancer are positive for HPV. However, a low percentage of HPV infected women ultimately develop low-grade CIN. Moreover, a low percentage of them progress towards high-grade CIN and invasive cervical carcinoma. Despite this, cervical cancer is an important public health problem, since 311,000 deaths per year are directly associated with this neoplasia [3]. In addition, 80% of these deaths occur in developing countries. High-risk zones of cervical cancer are South America, sub-Saharan Africa, and India. In the United States, there are more than 29,000 new cases per year including more than 11,000 deaths. The mortality rates are especially high in Chile and Mexico, while a low mortality rate is observed in Cuba, Puerto Rico, and Argentina [4,5]. On the other hand, head and neck cancer (HNC) is a heterogeneous group of tumors that arise in the head and neck region, which include the oral cavity, pharynx (e.g., nasopharynx, oropharynx, hypopharynx), paranasal sinuses, nasal cavity, larynx, and salivary glands [6]. HNCs are the sixth most frequent form of cancers worldwide, with an annual incidence of over 500,000 new cases [4]. Tobacco smoke (TS), alcohol consumption, and some viral infections, among other factors, are involved in its development [7]. HPV infection, in particular, has been etiologically associated with a subset of oropharyngeal and oral carcinomas [8]. When considering all HNCs, approximately 25% constitute HPV positive tumors, with a higher prevalence in oropharyngeal cancer (35.6%, range 11–100%) than in the oral cavity (23.5%, range 40–80%) or larynx (24%, range 0–100%) [9]. HPV is the main causal agent of the increased incidence of squamous cell carcinoma (SCC) of the oropharynx in developed countries, especially in males. The disease has a tendency to affect younger patients compared to those with HPV negative oropharyngeal carcinoma [10]. Epidemiologically, it is accepted that HPV positive oropharyngeal carcinomas are entities with a differential clinical behavior when compared to HPV negative cases [11]. 

## 2. Human Papillomavirus (HPV)

### 2.1. Structure, Classification and Replication Cycle

HPV is a small non-enveloped virus that belongs to the Papillomaviridae family. Its genome consists of circular double-stranded DNA that contains approximately 8000 bp organized in three regions: a regulatory, noncoding region known as the long control region (LCR), the early (E) region, which encodes for products involved in viral replication, and the late (L) region, which encodes for structural proteins [12]. The E region includes 6–8 open reading frames (ORF) that encode for E1, E2, E5, E6, E7, and E1^E4 proteins. L region encodes for L1 and L2 which are major and minor capsid proteins. The LCR is approximately 1000 bp in length and divided into three segments: a 5′ segment that contains transcription termination signals and a region of nuclear matrix binding; a central segment that contains binding sites for viral and cellular transcription factors involved in the regulation of the viral gene expression; and, finally, a 3′segment that contains the replication origin and the early promoter [13,14]. To date, more than 210 HPV genotypes have been described, characterized and classified into five genera (α, β, γ, μ, ν) based on the analysis of their genome sequence, tropism, and association with different diseases [15]. HPV enters basal cells through microlesions into the cutaneous or mucous epithelia. Then, viral particles through L1 protein bind to heparan sulfate proteoglycans (HSPGs), which promote a conformational change into the capsid. This allows them to bind to cell receptors involved in HPV entry [16,17]. Additionally, α6β4 integrins and tetraspanins are both required for HPV uptake [18,19]. Finally, virion endocytosis occurs through a macropinocytosis-like mechanism [20].

Once in the cytoplasm, the viral particle traffics in late endosomes where it dissembles, allowing L2 protein-mediated nucleus translocation [21]. In the nucleus, the viral genome remains as an episome attached to the host genome through the E2 viral protein where HPV replicates, usurping host replication and reparation machinery [22,23]. The HPV replicative cycle is closely related to the proliferation and differentiation of basal epithelial cells. E1 and E2 proteins, encoded by polycistronic RNAs, are involved in the replication and transcription of the viral genome. E1 is a helicase with ATPase activity that catalyzes unwinding DNA [24]. On the other hand, E2 protein works as a transcription factor that binds to cognate sequences in the LCR, regulating the expression of early transcripts in a dose-dependent manner [25]. It has been previously reported that high E2 concentrations prevent transcription factor binding, resulting in complete promoter repression. While at low concentrations, this works as a transcriptional activator of the early HPV promoter [26]. Therefore, the regulation of this promoter is critical for polycistronic RNA expression, which encodes for E6 and E7 oncoproteins. This ultimately leads to proteasomal degradation of p53 and pRb, which leads to the loss of ability to undergo apoptosis and uncontrolled proliferation, respectively [27,28,29]. Thus, both E6 and E7 are basally expressed to induce mitogenic and proliferative responses during viral replication in mucosal and cutaneous epithelia. Even though the roles of E6 and E7 have been well characterized, the role of E5 oncoprotein remains elusive [30]. E5 is located in the membrane of infected cells, where it binds to the epidermal growth factor receptor (EGFR), promoting its dimerization and activation, resulting in mitogenic signals to the nucleus, and thus stimulating cell proliferation [31]. Additionally, there is evidence that E5, E6, and E7 oncoproteins maintain an intracellular environment that promote the efficient replication of the viral genome with the involvement of E1 and E2 [32,33]. The replication of HPV concludes with late promoter (p670 in HPV16) activation in upper differentiated cells, which controls the L1 and L2 expressions. Finally, the assembly and maturation of new virions are carried out in upper layers of stratified epithelium [34]. Interestingly, the E4 protein is translated as an E1^E4 fusion protein from the E1^E4 mRNA [35], which is expressed in upper epithelial layers from the late promoter upstream of E1 ORF [36,37]. It has been demonstrated that calpain, a cellular cysteine protease, cleaves the HPV16 E1^E4 generating species able to multimerize to form amyloid-like fibers, ultimately disrupting the normal dynamics of the keratin networks in the upper epithelia layers [38]. This function of E1^E4 may facilitate the release of HPV particles.

### 2.2. Gene Expression Regulation

The regulation of HPV gene expression is complex and it involves viral and cellular factors, CpG methylation [39,40], alternative splicing (reviewed in [41]), and RNA polyadenylation [42]. Therefore, unregulated gene expression allows some HPV genotypes to establish long-term persistence in epithelial cells [43,44]. During the first stages of viral replication, the activity of the early promoter is regulated by cellular transcription factors. Among them, the Activator Protein-1 (AP-1), composed by the c-Jun and c-Fos family of proteins, promotes early viral gene expression during cell differentiation [45,46]. Interestingly, HR-HPV LCR contains binding sites for cellular transcription factors, among them AP-1 [47]. A wide range of cellular stimulus promotes AP-1 binding to DNA in specific sites called 12-O-tetradecanoylphorbol-13-acetate (TRE sites) to activate the expression of genes involved in cell proliferation, differentiation, apoptosis, and cell transformation [48,49]. On the other hand, Oct-1 [50,51] and TEF-1 are considered epithelia-specific regulators in some HPV genotypes [52]. Consequently, epithelium-specific transcription of HPV depends on a variety of cellular factors. However, each factor alone has no significant activity on the early promoter. Thus, a complex interplay among them is required to increase HPVs transcriptional activity [52]. This cooperation requires direct physical contact among different components as well as protein-protein interactions through regulation factors that do not establish direct contact with DNA [53]. Even though cell proteins play a key role in HPV transcription, some viral proteins are involved in the regulation of this process. The activity of E2 viral protein is critical, since it represses or activates the transcription depending on whether the corresponding binding site is close to the TATA box [54]. On the other hand, cutaneous HPV shows E2 binding sites (E2BSs) to less than 100 bp of the TATA box, which does not affect the transcriptional activity [55]. The E2 activation ability is a result of the amino-terminal end´s facility to physically interact with cell factors [25]. It has been demonstrated that E2 homodimer is able to bind to E2BSs, which consists of the palindromic consensus sequence 5′-ACCgNNNNcGGT-3′. In addition, E2 protein demonstrates an important function during viral DNA replication [56]. When E2 physically associates with E1 protein, E2 confers its sequence specificity for DNA binding. This interaction is necessary and sufficient to promote viral replication [57]. On the other hand, it has been described that E8^E2C protein is another important viral repressor of HR-HPV early promoter [58,59]. E8^E2C expression is critical for the maintenance of episomal forms of HPV in human foreskin keratinocytes [58]. Importantly, NCOR/SMRT repressor complexes interact with E8^E2C for efficiently inhibit viral replication [60].

HPV gene expression regulation does not only depend on transcription factors but also on post transcriptional events. During the viral replicative cycle, polycistronic RNAs are transcribed, which undergo alternative splicing through differential use of splicing sites located in the viral genome [41]. Thus, mRNAs expressed from the early promoter located at p97 (HPV16) are translated for expressing E6, E6*I, E7, E1, E2, E8, E4, and E5 proteins [41,61]. Interestingly, it was reported that E6*I transcripts facilitates E7 translation by increasing the space between the E6 stop codon and the E7 start codon [62].

During cell differentiation, the late promoter located at p670 (HPV16) is activated, promoting the high expression of L1, L2, E1^E4, and E5. Additionally, the E8 promoter is activated leading the expression of E8^E2C product [63]. It has been reported that E2 protein promotes late gene expression by inhibiting early polyadenylation [64]. In particular, it was suggested that E1^E4 fusion protein, by inhibiting Serine-Arginine (SR) Protein Kinase (SRPK1), regulates Serine-arginine (SR) protein-mediated host RNA metabolism and E2 nuclear localization [65]. Meanwhile, E6/E7 expression decreases, allowing for L1 and L2 expression. These conditions favor viral replication in the infected cell [43]. On the other hand, extracellular components are involved in HPV gene expression regulation. For instance, Chan et al. identified a nucleotide sequence-specific for glucocorticoid receptor binding in the HPV16 LCR [66]. Later, Yuan et al. demonstrated how progesterone enhances HPV16 E6 and E7 transcription in cell lines containing the viral genome [67]. 

### 2.3. HPV Role in Epithelial Tumors

Alpha (α)-HPVs are the genus with the best characterization because they have been implicated in cervical, anogenital and head and neck carcinogenesis [68]. In fact, in the 1970s, H. Zur Hausen detected HPV in warts and biopsies from women with cervical cancer [69]. Later, an etiological association between HPV infection and cervical cancer was confirmed [70]. Thus, according to their oncogenic potential, α-HPVs have been classified in high- and low-risk types. Low-risk (LR)-HPV genotypes such as HPV6 and 11 are associated with a majority of benign lesions such as condylomata acuminata and are involved in the development of respiratory papillomatosis. According to the International Agency for Research on Cancer (IARC), 12 high-risk (HR)-HPV types (16, 18, 31, 33, 35, 39, 45, 51, 52, 56, 58, and 59) are considered carcinogens. The HPV16 and 18 types are associated with approximately 70% of cervical cancer worldwide [70]. Interestingly, only HPV16 accounts for 90% of HPV positive head and neck SCCs [71,72]. The oncogenic mechanism of HR-HPV has been basically characterized in human keratinocytes. It has been reported that E6 and E7 oncoproteins interact with a significant number of intracellular proteins from cervical cancer-derived cells, leading to cancer promotion and progression [27]. Thus, E6/E7 overexpression is a necessary condition for HPV-mediated tumorigenesis, since it promotes cell proliferation. The HR-HPV full genomes often become integrated into the host chromosome, although the mechanisms involved in such integration remain elusive [73]. However, HPV-associated cancers contain integrated, episomal or episomal/integrated forms of HPV genome at variable number of copies [74,75]. In addition, it seems that integration events occur randomly in the host cell genome, even though specific hot spots have been reported in cervical exfoliated cells [76]. When integration is not detected in carcinomas, other mechanisms account for E6 and E7 overexpression [77,78,79]. For instance, frequent methylation in E2 binding site (E2BSs) motifs leading to E6/E7 upregulation in cervical cancer [77,80] and head and neck cancers has been described [81].

## 3. Tobacco Smoke and HPV Interactions

### 3.1. History and Epidemiology

As previously stated, additional factors are necessary for HPV-mediated carcinogenesis [69,82]. It has been proposed that some environmental chemical compounds alter HPV gene expression. In fact, in the first half of the 20th century, Peyton Rous discovered a cooperation between tar and papillomaviruses for inducing rabbit SCCs [83]. These historical experiments demonstrated that carcinogenesis may be induced by exposure to two different factors. In 1977, Winkelstein hypothesized that SCCs from different sites will be associated with TS. Therefore, considering that most cervical cancers are SCCs, it is expected that this tumor will be associated with TS [84]. In 1993, it was reported a possible cooperation between HPV16 and tobacco-related carcinogens for oral carcinogenesis. The authors demonstrated that primary oral keratinocytes immortalized with HPV16 acquire a tumor phenotype after treatment with nitrosamines and nitroguanosine, two carcinogens present in TS [85].

Epidemiological evidence strongly suggests a synergy between TS and HR-HPV infection for cervical cancer development [86,87,88]. In 2003, the IARC established that women who smoke are more susceptible to cervical cancer when compared to nonsmokers [89]. However, the mechanisms involved in such collaboration are unclear. According to the World Health Organization (WHO), TS is considered the most important human carcinogen involved in cancer initiation, promotion, and progression. Additionally, TS is considered the major cause of morbidity and mortality worldwide, causing more than five million deaths each year [90]. TS is a complex mixture of more than 4500 chemical compounds such as carbon monoxide (CO), hydrogen cyanide (HCN), nitrogen oxides (NOX), formaldehyde, acrolein, benzene, nicotine, nitrosamines, phenol/polycyclic aromatic hydrocarbons (PAH), and more [91]. Considering the established evidence that both TS and HPV are involved in epithelial cancer initiation, promotion, and progression, we speculate that a complex network of interactions exists. Due to the detection of TS metabolites in cervical mucous of women who smoke [92,93] and the direct exposure of oral cavity and nasopharynx to TS, the presence of both tobacco and HPV increases the possibility of direct interaction. Thus, considering molecular evidence, different findings have been reported (Table 1). 

### 3.2. Tobacco Smoke Affects HPV Replication

Alam et al. demonstrated that benzo[α]pyrene (BaP), a carcinogen present in TS, is able to increase the number of virions and genomes of HPV31 in organotypic epithelial cultures of cervical cells. The authors observed that high BaP concentrations resulted in a 10-fold increase in viral titers while low BaP concentrations increase the number of viral genomes [94]. It was suggested that this effect may increase the possibility of viral dissemination and persistence. Later, the same authors demonstrated that the Ras-Raf-Mek1/2-Erk1/2 signaling pathway is involved in the increase of HPV31 virions after BaP exposure. Moreover, Erk1/2 signaling promoted activation of CDK1 [95]. However, epidemiological studies did not find a dose-response relationship between TS frequency and HPV DNA load in cervical cancer [96,97,98]. In head and neck cancer HPV positive cases, studies reporting a TS dose-dependence respect viral load are lacking. More studies are warranted to explore this possibility.

### 3.3. Tobacco Smoke Promotes HPV E6 and E7 Expression

One study suggested that nicotine, an addictive component which is contained at high levels in TS, enhances the activity of HPV16 LCR synergistically with Brn-3a transcription factor and consequently promotes the expression of E6 and E7 oncoproteins [99]. Additionally, a recent report demonstrated that BaP increases the expression of HPV16 E7 in organotypic epithelial cultures of cervical cells. This expression is inhibited by curcumin, a potential chemopreventive natural compound [100]. On the other hand, Wei et al. demonstrated that acute exposure to cigarette smoke extracts (CSE) allows an increase in E6 and E7 when cervical cells maintain HR-HPV in an episomal physical status [101]. In the same way, Peña et al. reported that CSE promote HPV16 p97 promoter activation in lung epithelial cells, resulting in increased E6/E7 levels [102]. The mechanism involved in TS-mediated E6 and E7 expression is unclear, although it seems that AP-1 may be involved [103]. The activity of AP-1 can be regulated by c-Jun and c-Fos phosphorylation through MAPK pathway activation in response to diverse stimuli (i.e., exposure to TS) [14,104]. Interestingly, it was reported that CSE enhances AP-1 activity, which in turn up-regulates proangiogenic cytokines in oropharyngeal cells [105]. In addition, carcinogens from TS promote mutations that favor AP-1 binding to HPV16 LCR, increasing the activity of p97 promoter [106,107]. Moreover, evidence indicates that TS promotes AP-1 activity in several cell types [108,109,110]. In lung cells, transcription factors, such as NF-κB, HIF, and AP-1, have been shown to be highly sensitive to CSE [111,112], playing a critical role in the inflammatory response due to injury induced by reactive oxygen species (ROS) [113]. In this scenario, AP-1 activity can be altered by the induction of kinases that respond to the dimerization and phosphorylation of tyrosine kinases receptors. In fact, in the TS-induced lung pathogenesis, the EGFR was found to be overexpressed and aberrantly activated by TS [114]. The classical pathway induced by TS is MAPK, which through multiple factors, mainly ERK1/2, can direct the signal to the nucleus to promote AP-1 phosphorylation, promoting their binding to cognate regulatory elements in cell promoters [115,116,117]. Moreover, TS has been demonstrated to induce upregulation of c-Jun, c-Fos, and Fra-1, but not of Fra-2, Jun-B, and Jun-D expression [109,118], suggesting an obligatory role for EGFR-mediated MAPK activation. Evidence also reveals that TS induces the expression of cyclin D1 and PCNA, which are AP-1 targets genes involved in promoting cell cycle progression [108]. Likewise, in human bronchial cells, it has been shown that TS induces increased AP-1 activity, promoting aberrant c-Jun/Fra1 dimerization, which is linked with aberrant phenotypical changes such as hyperplasia, epithelial-mesenchymal transition (EMT), release of EGF ligand, and cell transformation [119,120]. Additionally, AP-1 activity is regulated by TS at different levels through its phosphorylation, dimer composition, and mRNA expression [103]. Taken together, it is plausible that AP-1 is involved in TS-mediated E6 and E7 overexpression.

### 3.4. Tobacco Smoke Affects Immune Responses Against HPV

It has been established that TS is associated with early cervical carcinogenic alterations and a decreased ability of the immune system to induce HPV clearance in the cervix [121,122]. In fact, TS affects both innate and adaptive immune responses [123]. One study reported that NK cell activity was significantly decreased in subjects who smoked [124,125]. On the other hand, the same study reported a decreased amount of serum antibodies in subjects who smoked, explaining, at least in part, a higher susceptibility to infections. In addition, a reduction of Langerhans cells and helper T cells in the uterine cervix transformation zone for women who smoke was reported, suggesting a decreased local immunosurveillance [126]. Moreover, a reduction of smoking was associated with changes in immune cell counts [127]. It was further demonstrated that TS shows direct immunosuppressive effects on T cells, increasing the percentage of CD8+ T cells while lowering CD4+ T cells [128]. 

Regarding specific compounds, nicotine shows immunosuppressive properties in some animal models [129,130]. Moreover, nicotine is able to promote migration in cervical cancer cells by activating the PI3K/Akt/NF-κB pathway, which suggests connections between TS and cervical cancer progression [131]. Interestingly, acrolein, a major component of TS, it was reported to suppress the inflammatory and innate response by reducing levels of IL-8 mRNA and human beta-defensin (HBD) in sinonasal epithelial cells [132]. Alike, BaP, at low-dose, was demonstrated to have immunosuppressive effects in activated murine marrow-derived macrophages in an aryl hydrocarbon receptor (AhR)-dependent manner [133]. Taken together, studies support immunosuppressive effects of TS and/or some specific compounds such as nicotine, BaP, and acrolein, which may favor increased infection susceptibility and/or impairing its clearance. 

### 3.5. Tobacco Smoke Promotes DNA Damage Leading to an Increased HPV Oncogenic Role

The most important molecular alterations caused by TS are adduct generation and the activation of signaling pathways after binding tobacco components (i.e., nicotine) to membrane receptors [134]. Some compounds present in TS are processed and metabolically activated by oxidoreductases, such as cytochrome P450, after which hydrosoluble electrophilic molecules bind to the genome leading the production of covalent and stable complexes with nitrogenated bases, known as DNA adducts [135,136] which can lead to mutations and cancer [137]. 

It has been demonstrated that HPV16 and 18 infected cells are capable of generating high levels of BaP metabolites, thus increasing adduct formation, DNA damage and probably, favoring the possibility of HPV integration into the host [138]. In accordance with this finding, HPV-transformed cervical cancer cells were highly susceptible to DNA damage promoted by CSE, detected by comet assay [139]. Interestingly, Wei et al. found increased DNA mutations and double strand breaks in CIN I cells harboring episomal HPV after TS exposure [101]. Moreover, the authors showed that TS increases p53 levels in normal HPV-negative cervical cells, thus activating DNA repair and apoptosis. On the contrary, in HR-HPV-infected cells, TS induces a decrease in p53 activity by promoting overexpression of viral oncoproteins. The authors finally suggested that cervical cells harboring episomal HPV are more susceptible to DNA damage promoted by TS, when compared to those cells harboring integrated HPV, concluding a more pronounced effect in early lesions [101]. In line with this finding, an article revealed higher levels of p53 in both cervical cancer cells and HPV (E6E7) - transfected cells exposed to TS. Additionally, p53 expression was not altered in HPV-negative (C33A) human cervical cells exposed to TS for 72 h. These findings suggest the presence of residual p53 activity in HPV positive cervical carcinoma cells [140,141].

It is known that TP53 is frequently mutated in HPV negative tumors, such as those associated with TS [142,143]. Specifically, missense TP53 mutations frequently result in p53 accumulation in the nucleus of tumor cells [144]. Regarding this issue, an in vivo study in head and neck carcinoma samples from 110 patients showed that tobacco consumption was highly associated with p53 staining intensity on immunohistochemistry [145]. On the contrary, TP53 is not frequently mutated in HPV-driven tumors, even though p53 levels are significantly decreased by HR-HPV E6 oncoprotein. Thus, in non-tumor HPV infected cells, TS exposure promotes functional p53 activity, which is abrogated by E6 oncoprotein, leading to increased DNA damage and cancer initiation [101]. On the other hand, Muñoz et al. found that TS and E6/E7 collaborates for increasing tumor properties of lung epithelial cells [146]. Moreover, Peña et al. showed that E6 and E7 cooperate with TS for increasing DNA damage in tumor epithelial cells [102]. Another study reported a higher sensitivity to the inactivation of clonogenic survival to BaP in HR-HPV E6, expressing fibroblasts compared to normal and inactive-p53 lines (p53-H179Q and p53-RNAi) exposed to the same cigarette smoke carcinogen. Moreover, it was proposed that HR-HPV E6 may activate G2 checkpoint kinase CHK1, a well-known DNA damage marker, in fibroblasts exposed to BaP. Altogether, all of these data suggest a synergistic interaction between TS and HPV for increasing the DNA damage in epithelial cells, ultimately leading to cancer initiation or progression (Figure 1) [141].

### 3.6. Tobacco Smoke Alters Cellular Gene Expression Involved in HPV-Mediated Carcinogenesis

Due to the strong relationship between TS and lung cancer, the role of tobacco has been highly studied in lung cells and patients with this disease. In fact, it has been demonstrated that lung cells exposed to CSE show gene expression alterations in around 3700 genes, the cytochrome P450 enzyme among them [137]. In lung cells, the exposure to CSE is associated with chromatin remodeling, which, in turn, leads to differential gene expression involved in oxidative stress responses. Interestingly, these effects are enhanced in patients with viral infections [147].

As previously mentioned, viral oncogenes E6 and E7 play a crucial role interfering with pRb and p53 tumor suppressor functions, which are important events in HPV-mediated carcinogenesis [148]. Some studies have assessed the interplay between TS and HPV in such gene expression. The effect of HPV and TS on p53 was addressed in Section 3.5. On the other hand, Alam et al. evaluated the effect of BaP on tumor suppressor pRb and other proteins expressions in HPV31b+ cervical intraepithelial neoplasia culture. They observed that inactive pRb (hyperphosphorylated) form was significantly upregulated compared to the negative control, proposing that the regulation by BaP may not be correlated with the HPV E7 effect. Interestingly, CDK1, a protein active in the G2 phase of the cell cycle, exhibited a significant increase in HPV31b raft cultures treated with BaP. The authors suggested that this carcinogenic agent may modulate CDK1 kinase activity in HPV-positive in order to favor cancer progression [149]. This may be supported by the evidence that E6 could upregulate CDK1 through E2F1, shedding light on a promising mechanism in which HPV induces genomic instability [150]. 

Another study demonstrated that eukaryotic translation initiation factor 4E (eIF4E), a crucial factor in translation control and recently evaluated as a plausible oncogene, is induced by E6 and E7 proteins. It has been shown to influence cell proliferation, migration, and apoptosis [151,152]. Moreover, the contribution between nicotine and HPV was assessed in another study. This article revealed that nicotine upregulates eIF4E expression in HPV-immortalized cervical epithelial (H8) cells, and subsequently enhances the expression of oncogenes c-Myc, VEGF, Cyclin D1, and Bcl-2 expressions [153]. In addition, the same researchers postulated a novel mechanism wherein nicotine may promote cell proliferation in HPV16-positive human cervical epithelium cells. In nicotine-treated cells, they observed that the reduction of small ribosomal protein RPS27 expression induced the phosphorylation of ubiquitin-protein ligase Mdm2 and consequently lead to p53 reduction [154]. Moreover, it has been documented that HPV E6 oncoprotein enhances the acknowledged Wnt/β-catenin pathway by inducing FOXM expression through the MZF1/NKX2-1 axis and, consequently, it overexpresses target genes such as Cyclin D and c-Myc. These results may explain a new mechanism whereby HPV may promote invasiveness and stemness in oral and lung cancer cells [155]. Curiously, these findings differ from a recent study in HPV-positive oral carcinoma cells from smoking patients, in which the authors found that the Wnt/βCatenin pathway and target genes (Cyclin D1, Cdh1, Cdkn2a, Cd44, Axin2, c-Myc, and Tcf1) are downregulated in HPV-positive oral cells compared to those HPV-negative. These unexpected results suggest a particular pathological form in HPV-positive oral cancers associated with TS [156]. However, as this was a comparative gene expression study in two oral cell lines, HPV-mediated Wnt/βCatenin activation cannot be denied. 

Gene expression alterations in HPV-driven tumorigenesis have been reported which may also be altered by TS. A compelling example of this is the *PIR* gene, which encodes the redox sensor Pirin and is upregulated by TS in bronchial cells [157]. Likewise, it was observed that *PIR* is upregulated by HR-HPV E7 oncoprotein dependent on EGFR/MEK/ERK and PI3K/Akt signaling pathways. Subsequently, Pirin protein induces NF-κB activation and is involved in EMT and migration in HPV-positive cervical cancer cells [158]. Taken together, these data indicate that *PIR* may be important in both TS and HPV-mediated carcinogenesis. A similar role may apply to the redox-sensitive transcription factor NRF2, which is induced by TS and also upregulated in HPV16 positive cervical cancer cells, suggesting it may act as a potential oncogene [159,160]. Moreover, CXCL14 is a chemotactic factor for dendritic cells, NK cells, and an angiogenesis inhibitor, and it is downregulated in HPV-positive cervical cells and tissues in an E7-dependent manner. Furthermore, oncoprotein E7 triggers CXCL14 promoter hypermethylation. Therefore, these findings suggest a new HPV-mediated oncogenesis route by suppressing antitumor CXCL14 effects [161]. Surprisingly, CXCL14 expression is also highly downregulated in human bronchial epithelial cells exposed to CSE [162]. Altogether, this indicates a common tentative oncogenesis-related factor. In the same way, CXCL12/CXCR4 pathway induces HPV-positive keratinocytes transformation due to high levels of oncoprotein-induced signaling pathways and cell proliferation correlation [163]. Nicotine also stimulates this mechanism by overexpressing CXCR4, thus playing a key role in tumor progression [164]. PTPN14, another candidate gene implicated in metastasis inhibition and negative oncogenesis regulation, has been recently reported as an important mediator in HPV-carcinogenesis. These observations show that HPV16 E7 degrades PTPN14 to inhibit keratinocyte differentiation independent of RB1 inactivation [165]. Another study detected hypophosphorylation of PTPN14 in the presence of TS [166], demonstrating a potential function in HPV-positive cancers. Altogether, a plethora of genes have been shown to be altered in the presence of both TS and HPV, suggesting the possibility of cooperation.

## 4. Conclusions and Remarks

HR-HPV infection is a necessary although not sufficient condition for cervical carcinogenesis. Women who smoke are more susceptible to this disease, suggesting that TS is a cofactor. Additionally, although HPV-driven HNCs are a clinical entity distinct from those HPV-negative cases, a portion of HPV-driven HNCs are from subjects who smoke.

Diverse mechanisms have been suggested for a collaboration between HPV and TS in epithelial cancers. First, both TS and HR-HPV affect a plethora of signaling pathways involved in cancer initiation, promotion, and progression, suggesting a complex network of interactions. In addition, experimental approaches demonstrated that TS exposure results in increased E6 and E7 expression and DNA damage in epithelial cells. Moreover, TS affects both innate and adaptative immune responses against HPV. Additionally, specific compounds such as BaP increase HPV replication in cervical cancer cells, with acrolein or nicotine showing immunosuppressive properties. We cannot deny the possibility that additional mechanisms that remain to be discovered may be involved in TS/HPV interactions in cervical and HNCs.

Due to the fact that better survival and outcomes in HPV-positive HNCs has been established, it will be of interest to assess the clinical consequences of TS/HPV interaction in patients with HPV-driven tumors. In addition, other environmental factors or pathogens may be involved in collaboration with HR-HPV for epithelial carcinogenesis. The knowledge of such factors and mechanisms will be important to establish both prevention strategies and to find new therapeutic targets for cervical or HNC treatment. Finally, oncogenic viruses such as HPV are good models to study interactions with environmental or host-related factors for human carcinogenesis.

## Figures and Tables

**Figure 1 cancers-12-02201-f001:**
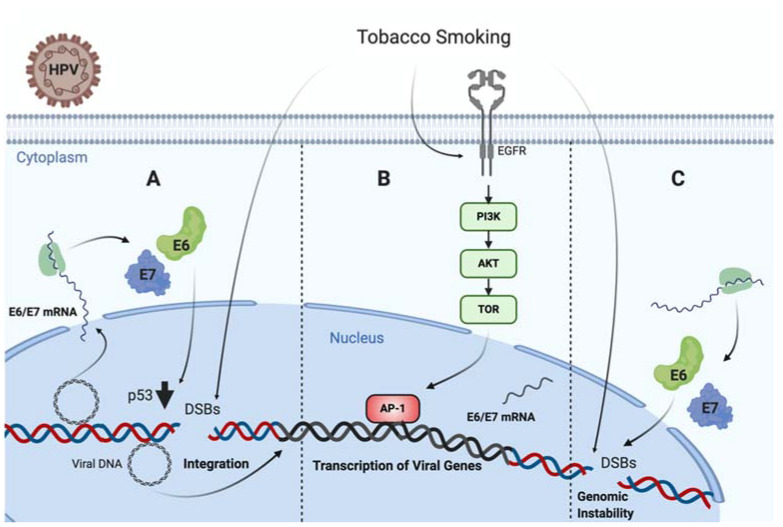
Tobacco smoke cooperates with high-risk human papillomavirus (HR-HPV) for increased DNA damage in epithelial cells. (**A**) Tobacco smoke causes E6/E7 oveerexpression and DNA damage in cells harboring HPV episomal forms, inducing p53 downregulation and potentially promoting viral genome integration [101]. (**B**) Tobacco smoke promotes EGFR/PI3K/AKT activation inducing AP-1 recruitment to the LCR and activating the HR-HPV early promoter, thus increasing E6/E7 expression [103]. (**C**) Both tobacco smoke and E6/E7 oncoproteins cooperate for increasing genomic instability in infected cells [101,138].

**Table 1 cancers-12-02201-t001:** Tobacco smoke promotes viral and host-related changes involved in epithelial carcinogenesis ^1^.

Molecule	Effect	Cell Line	Ref.
**Tobacco Smoke Affects HPV Replication**			
BaP	Increases number of virions and genomes of HPV31	CIN (CIN-612 9E)	[94]
BaP	Increases number of virions of HPV31 by MAPK ERK1/2 pathway	CIN (CIN-612 9E)	[95]
**Tobacco Smoke Promotes HPV E6/E7 Expression**			
Nicotine	Promotes indirectly HPV16 E6 and E7 expression	CIN1, CIN2, CIN3	[99]
		and cervical cancer	
BaP	Increases HPV16 E7 expression	Cervical carcinoma (CaSki)	[100]
CSE	Increases E6 and E7 expression in cells maintaining episomal HPV16 genomes	CIN1	[101]
		(W12 and CIN612)	
CSE	Increases tumor properties in cells expressing HPV16 E6 and E7	Lung adenocarcinoma (A549)	[146]
		Lung carcinoma (A549),	
CSE	Induces HPV16 p97 promoter activation	bronquial carcinoma (H-2170),	[102]
		bronquial (BEAS-2B), alveolar (NL-20),	
		cervical carcinoma (CaSki and SiHa)	
CSE	Promotes mutation that favors AP-1 binding to HPV16 LCR, increasing the activity of p97 promoter	Oral keratinocyte	[106]
		(16BNNK)	
CSE	Increases HPV16 E6 and E7 through p97 promoter activation involving EGFR/PI3K/Akt/C-Jun signaling pathway	Cervical carcinoma	[103]
		(CaSki and SiHa)	
**Tobacco Smoke Promotes DNA Damage**			
Nicotine	Generates adducts and actives Ras signaling pathway	Lung epithelial (LA4)	[134]
Acrolein	Remodels chromatin leading to oxidative stress responses	Bronchial epithelial (BEAS-2B),	[147]
		Lung adenocarcinoma (A549)	
CSE	Increases DNA damage in cells maintaining episomal HPV16 genomes	CIN1	[101]
		(W12 and CIN612)	
**Tobacco Smoke Affects Immune Responses**			
Nicotine	Induces T cell anergy	T cells	[129]
Nicotine	Promotes migration in HPV18 cells by activating	Cervical carcinoma	[131]
	the PI3K/Akt/NF-κB pathway	(HeLa)	

^1^ Abbreviations: CSE: cigarette smoke extracts; CIN: cervical intraepithelial neoplasia; BaP: benzo[α]pyrene.

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
