# Peer review of "High-Risk Human Papillomavirus and Tobacco Smoke Interactions in Epithelial Carcinogenesis"

_cancers, 2020, doi:10.3390/cancers12082201_

Round 1
Reviewer 1 Report
In this manuscript, “High-risk human papillomavirus and tobacco smoke interactions in epithelial carcinogenesis”, the authors aim to catalog the molecular mechanisms between high-risk HPV and tobacco smoke that contribute to carcinogenesis (primarily head and neck and cervical). The authors provide a thorough background and history of HPV as well as molecular pathway interactions with tobacco smoke.
The main strength of the paper lies in the comprehensive descriptions of how tobacco smoke alters the consequences of hrHPV in the context of carcinogenesis. Cellular/ viral molecular relationships and the impact of how they are altered by tobacco smoke at viral replication and expression, immune response, DNA damage, and cellular gene expression are all thoroughly explained with up-to-date references cited. I do not see any major areas of weakness in this paper. One aspect that needs to be added is the contribution of tobacco smoke to the rate of viral integration into the cellular genome, and how viral integration is influenced by carcinogens in tobacco smoke.
Other minor issue: Figure one is incredibly simplistic and really does not add any information.
Author Response
Many thanks for this comment. We added sentences related to viral integration after DNA damage caused by HPV. Although the increase of DNA damage by tobacco smoke, suggest increased integration, according to our knowledge, this possibility has not been directly explored by in vitro approaches. In addition, the Figure 1 was deleted.
Reviewer 2 Report
This manuscript reviews a cooperative interaction between the high risk human papillomavirus (HPV) and tobacco smoke (TS) for epithelial carcinogenesis. The topic is generally important; however, the content of this manuscript requires improvements.
The figures of this review article are too simple and are not informative to illustrate the content. Instead, use of Table that lists the effect of TS (including specific components) on HPV activation, replication, and tumorigenicity through AP-1 (RTK-MAPK), immunomodulation, and other signaling will be a more comprehensive way to explain these complicated effects.
Many studies mentioned in the section 3, Tobacco smoke and HPV interaction, are association observation, so please propose, based on these observational studies, the causative links between TS and HPV to get more deep insight to the carcinogenic mechanism. In addition, please discuss possible reasons regarding some conflict results from different studies, such as the effect of TS/HPV on p53, WNT signaling (section 3.6).
Author Response
Many thanks for these comments. A Table including TS and HPV interactions was included. The Figure 1 was deleted, and the Figure 3 was significantly improved to be more informative. In addition, some conflicting results were discussed.
Reviewer 3 Report
This is a well-conceptualized review by Aguayo et al. aimed to summarize what is known about tobacco smoke interactions in HPV-induced epithelial cancers. Overall, the review captures many/most important scholarly details of HPV disease, HPV replication and gene expression, tobacco constituent interactions with HPV, and how the two alter normal cell functions. These aspects cannot be over-stated as important. However, there are limitations of this review that should be revised to greatly improve upon the understanding and impact of this work.
One main limitation of the work is that the language needs improvement. There are a number of issues with word usage, incorrect wording and grammar. The work would benefit tremendously by editing from a native English speaker. For example, the word “genders” is used when the authors mean “genera.” In line 134, the phrase “suffering alternative splicing” is not clear (suffering is not the correct word). The authors often use the Spanish “y”, which should be converted to the English form, “and”. “Smoker women” are better described as “women smokers”.
There are some important aspects of HPV regulation that are overlooked. Three specific examples include information about E6*, E8^E2C, and E1^E4. The E6* splice variant appears to play a role in E7 protein expression and E8^E2C is a negative regulator of E2 protein functions. The authors do not make clear that the E1^E4 mRNA is highly expressed from the early promoter, but the E4 protein is technically a late gene product that plays known roles in the “late” phase of viral replication and virion morphogenesis. The concept of viral genome integration is oversimplified. The important point is that transformation by HPV requires E6/E7 expression. The integration of HPV genome integration into host chromosomes during carcinogenesis is not requisite. However, when integration occurs, E6/E7 gene expression is always maintained. When integration is not detected, again over-expression of E6/E7 is necessary for the transformed phenotype.
In section 2.2, (line 112) the correct designation of AP-1 is “Activator Protein” not “Associated Protein”. In section 3.3, the text from lines 223-230 is a bit redundant with prior section 2.2 and could be tied into this section more concisely. This information, including “AP-1 binding to DNA in specific sites called 12-O-tetradecanoylphorbol-13-acetate (TRE sites) to activate the expression of genes involved in cell proliferation, differentiation, apoptosis and cell transformation [71,72].” noted in section 2.2. it might be helpful to separate the regulation of HPV gene expression by viral factors (E2, E8^E2C) from the regulation by cellular factors (AP-1, p53, etc).
Some studies appear to be summarized in more detail than others. It is not always clear that the details of certain studies are needed to capture the overall intent of the review, and the authors might reevaluate the need for detail in each case.
Other studies and their findings are not completely captured in the full story. For example, in section 3.5, Wei et al. also showed that TS increases DNA strand breaks in HPV infected cells, and that (presumably) the increase in E6/E7 activities, prevented DNA damage repair and allowed the survival of cells with mutations compared to HPV-negative cells. In this same section, it is not clear how the text beginning in line 283 (MAPK) fit into the DNA damage theme of this paragraph.
Some studies may not be accurately represented. For example, beginning in line 176, the hypothesis posed by Winkelstein does not appear to be appropriately captured. Citing other review articles should be minimized unless discussing broad, well-known concepts. In particular, the statement in lines 228-230 “AP-1 is a key transcription factor during the HPV replication cycle since its expression during cell differentiation promotes and maintains E6 and E7 expression in stratified epithelia [23].” To this reviewers’ knowledge, this statement is not a clearly tested idea and is simply a potential explanation speculated in another review. There may well be other primary studies that show this, but they should be cited rather than a review.
Figure 1 and the legend are not overly informative as presented. HPV is an initiating event and E6/E7 expression must be maintained, but the virus replicative cycle per se, is not necessarily maintained (as might be indicated by the virus particle).
Figure 2 is rather simplistic and could be more informative by including how tobacco constituents activate EGFR to both cause proliferation and dampens the immune responses. E5, E6, E7 actions can feed into activating EGFR, which creates causing a feed-forward activation loop.
Figure 3 could be broken down so that the functions of HPV replication and E6/E7 that also alter immune responses, DNA damage response, and host gene expression are also, separately captured.
Some jargon or words less familiar to wider audiences could be briefly explained. Examples are p105 (presumably a pRb family protein, noted in line 122) and “raft” (lines 202, 214, 317), which refers to organotypic epithelial tissue models of neoplasia.
It is not advisable to cite a manuscript in preparation, but rather the citation should be included as such in the text (ref. 117).
Author Response
REFEREE 3.- This is a well-conceptualized review by Aguayo et al. aimed to summarize what is known about tobacco smoke interactions in HPV-induced epithelial cancers. Overall, the review captures many/most important scholarly details of HPV disease, HPV replication and gene expression, tobacco constituent interactions with HPV, and how the two alter normal cell functions. These aspects cannot be over-stated as important. However, there are limitations of this review that should be revised to greatly improve upon the understanding and impact of this work.
One main limitation of the work is that the language needs improvement. There are a number of issues with word usage, incorrect wording and grammar. The work would benefit tremendously by editing from a native English speaker. For example, the word “genders” is used when the authors mean “genera.” In line 134, the phrase “suffering alternative splicing” is not clear (suffering is not the correct word). The authors often use the Spanish “y”, which should be converted to the English form, “and”. “Smoker women” are better described as “women smokers”.
ANSWER. Many thanks for these comments. The manuscript was extensively reviewed by a native English speaker, so these mistakes were corrected.
REFEREE 3.- There are some important aspects of HPV regulation that are overlooked. Three specific examples include information about E6*, E8^E2C, and E1^E4. The E6* splice variant appears to play a role in E7 protein expression and E8^E2C is a negative regulator of E2 protein functions. The authors do not make clear that the E1^E4 mRNA is highly expressed from the early promoter, but the E4 protein is technically a late gene product that plays known roles in the “late” phase of viral replication and virion morphogenesis.
ANSWER. Many thanks for these observations. These very important aspects related to regulation of HPV gene expression were included in the manuscript (section 3.2). The corresponding references were added.
REFEREE 3.- The concept of viral genome integration is oversimplified. The important point is that transformation by HPV requires E6/E7 expression. The integration of HPV genome integration into host chromosomes during carcinogenesis is not requisite. However, when integration occurs, E6/E7 gene expression is always maintained. When integration is not detected, again over-expression of E6/E7 is necessary for the transformed phenotype.
ANSWER. Many thanks for these comments. We absolutely agree with these descriptions, so the manuscript was improved to clarify these aspects (section 3.3) and references were included.
REFEREE 3.- In section 2.2, (line 112) the correct designation of AP-1 is “Activator Protein” not “Associated Protein”.
ANSWER: It was corrected.
REFEREE 3.- In section 3.3, the text from lines 223-230 is a bit redundant with prior section 2.2 and could be tied into this section more concisely. This information, including “AP-1 binding to DNA in specific sites called 12-O-tetradecanoylphorbol-13-acetate (TRE sites) to activate the expression of genes involved in cell proliferation, differentiation, apoptosis and cell transformation [71,72].” noted in section 2.2. it might be helpful to separate the regulation of HPV gene expression by viral factors (E2, E8^E2C) from the regulation by cellular factors (AP-1, p53, etc).
ANSWER: Many thanks for this suggestion. These aspects were corrected.
REFEREE 3.- Some studies appear to be summarized in more detail than others. It is not always clear that the details of certain studies are needed to capture the overall intent of the review, and the authors might reevaluate the need for detail in each case.
ANSWER: The manuscript was checked, some aspects such as Immune responses or DNA damage were enriched with more relevant information and additional significant aspects from some papers, were added (ie. Alam S; Wei et al; etc).
REFEREE 3.- Other studies and their findings are not completely captured in the full story. For example, in section 3.5, Wei et al. also showed that TS increases DNA strand breaks in HPV infected cells, and that (presumably) the increase in E6/E7 activities, prevented DNA damage repair and allowed the survival of cells with mutations compared to HPV-negative cells. In this same section, it is not clear how the text beginning in line 283 (MAPK) fit into the DNA damage theme of this paragraph.
ANSWER: We included in the manuscript additional findings by Wei et al. In addition, additional references were added. The MAPK sentences were deleted from this section.
REFEREE 3.- Some studies may not be accurately represented. For example, beginning in line 176, the hypothesis posed by Winkelstein does not appear to be appropriately captured. Citing other review articles should be minimized unless discussing broad, well-known concepts. In particular, the statement in lines 228-230 “AP-1 is a key transcription factor during the HPV replication cycle since its expression during cell differentiation promotes and maintains E6 and E7 expression in stratified epithelia [23].” To this reviewers’ knowledge, this statement is not a clearly tested idea and is simply a potential explanation speculated in another review. There may well be other primary studies that show this, but they should be cited rather than a review.
ANSWER: Many thanks for this observation. Reviews were replaced or complemented by primary original articles in the manuscript. The manuscript was completely checked.
REFEREE 3.- Figure 1 and the legend are not overly informative as presented. HPV is an initiating event and E6/E7 expression must be maintained, but the virus replicative cycle per se, is not necessarily maintained (as might be indicated by the virus particle).
ANSWER: The Figure 1 was deleted from the manuscript. We included a table summarizing findings related to interactions between HPV and tobacco. In addition, the final model was modified.
REFEREE 3.- Figure 2 is rather simplistic and could be more informative by including how tobacco constituents activate EGFR to both cause proliferation and dampens the immune responses. E5, E6, E7 actions can feed into activating EGFR, which creates causing a feed-forward activation loop.
ANSWER: The Figure 2 was deleted, and this information was added to the Figure 3.
REFEREE 3.- Figure 3 could be broken down so that the functions of HPV replication and E6/E7 that also alter immune responses, DNA damage response, and host gene expression are also, separately captured.
ANSWER: The figure 3 was replaced by another in which different aspects of tobacco smoke and HPV interactions are addressed. In addition, a Table was included summarizing HPV/TS interactions.
REFEREE 3.- Some jargon or words less familiar to wider audiences could be briefly explained. Examples are p105 (presumably a pRb family protein, noted in line 122) and “raft” (lines 202, 214, 317), which refers to organotypic epithelial tissue models of neoplasia.
It is not advisable to cite a manuscript in preparation, but rather the citation should be included as such in the text (ref. 117).
ANSWER: Many thanks for these observations. These words were replaced by other more known expressions. The “Manuscript in preparation” was deleted.
Reviewer 4 Report
Aguayo et. al., have written a very comprehensive review of the potential link between tobacco smoke (TS) and Human Papillomavirus (HPV) on cancers of epithelial origin. The reviewers covered literature covering genital and oral carcinomas. The background on HPV biology (genes/etc) was very well covered as was the literature of the various alterations of gene expression that are theorized to drive cells into a transformed phenotype. The review stops short of detailing mechanisms of the various mutations, which is reasonable, but do a good job of describing the role of the viral E2, E5, E6, and E7 proteins.
The review article is easy to read, and will be a great tool for the field of HPV induced oncogenesis, especially when linking it to TS.

Author Response
Many thanks for these comments.
Round 2
Reviewer 2 Report
The authors have answered my questions and the manuscript has been greatly improved. However, inappropriate allocation of some studies in Table 1 should be corrected before this manuscript can be published, such as nicotine-induced T-cell anergy should be in the category of immune response; and the two studies listed in the category of immune response are more related to the category of DNA damage.
Author Response
Many thanks for these observations. Table 1 was corrected.
Reviewer 3 Report
The English is much improved but there remain issues with grammar and some odd word usages. I have annotated the pdf with specific comments. When sections of text are circled without a note, it signals that the grammar is incorrect.
The review is unnecessarily long and complicated and some sections are not organized well (they jump around and could be streamlined). Some less relevant topics are still considered in more detail than are relevant to TS. The findings of studies could be summarized with fewer specific details of the experimental designs or techniques used. Too much time is spent describing early and late gene products that are not relevant to HPV transformation. The conclusion is rather brief and gives no sense of the overall message, where the field is going or what needs to be examined. Are there markers of TS+HPV cancers that will need special therapies?
There is little discussion of Table 1, which seems like it should be the focus of the paper.
Fig. 1 needs more explanation. There is no accounting for the fact that TS activates E6 and E7 before integration.
Some statements remain obscure or misleading.

Author Response
Reviewer. The English is much improved but there remain issues with grammar and some odd word usages. I have annotated the pdf with specific comments. When sections of text are circled without a note, it signals that the grammar is incorrect.
Answer. Many thanks for this observation. As previously appointed, the manuscript was checked by a native English speaker from MDPI. We corrected the additional observations respect English grammar.
Reviewer.The review is unnecessarily long and complicated and some sections are not organized well (they jump around and could be streamlined). Some less relevant topics are still considered in more detail than are relevant to TS. The findings of studies could be summarized with fewer specific details of the experimental designs or techniques used. Too much time is spent describing early and late gene products that are not relevant to HPV transformation.
Answer. Many thanks for the exhaustive revision; we addressed and corrected all of these observations submitted (pdf file) by this reviewer. In addition, some specific sentences were deleted (i.e. related to AP-1) because may be considered irrelevant for this manuscript.
Reviewer.The conclusion is rather brief and gives no sense of the overall message, where the field is going or what needs to be examined. Are there markers of TS+HPV cancers that will need special therapies?
Answer. More sentences were added to the conclusion, summarizing the suggested the tobacco smoke/HPV interactions; some perspectives and suggestions on future research were added.
Reviewer. There is little discussion of Table 1, which seems like it should be the focus of the paper.
Answer. The Figure 1 summarize experimental findings related to tobacco smoke /HPV interactions. Thus, the topics included in the Figure 1 were discussed in the manuscript.
Reviewer 1. Fig. 1 needs more explanation. There is no accounting for the fact that TS activates E6 and E7 before integration.
Answer. Figure 1 is a scheme summarizing some mechanistic findings related to tobacco smoke/HPV interactions. Section A represents the findings by Wei et al. (2014). As we can observe, in the presence of tobacco smoke, episomal forms of HPV overexpress E6/E7 promoting p53 downregulation and increased DNA damage, probably increasing HPV genomes integration. Section B summarizes the findings by Muñoz et al (2018). Section C summarizes additional findings by Wei et al. (2014) and others. The references were added in the legend.
Reviewer 1. Some statements remain obscure or misleading.
Answer. The manuscript was completely checked and corrected according to the suggested corrections.